# Productive Performance and Cecum Microbiota Analysis of Broiler Chickens Supplemented with β-Mannanases and Bacteriophages—A Pilot Study

**DOI:** 10.3390/ani12020169

**Published:** 2022-01-11

**Authors:** Carlos A. Pantoja-Don Juan, Gabriela Gómez-Verduzco, Claudia C. Márquez-Mota, Guillermo Téllez-Isaías, Young M. Kwon, Arturo Cortés-Cuevas, José Arce-Menocal, Daniel Martínez-Gómez, Ernesto Ávila-González

**Affiliations:** 1Departamento de Medicina y Zootecnia de Aves, Facultad de Medicina Veterinaria y Zootecnia, Universidad Nacional Autónoma de México, Avenida Universidad 3000, Ciudad de México 04510, Mexico; pantoja.9106@gmail.com; 2Departamento de Nutrición Animal y Bioquímica, Facultad de Medicina Veterinaria y Zootecnia, Universidad Nacional Autónoma de México, Avenida Universidad 3000, Ciudad de México 04510, Mexico; 3Department of Poultry Science, University of Arkansas, Fayetteville, AR 72701, USA; gtellez@uark.edu (G.T.-I.); ykwon@uark.edu (Y.M.K.); 4Centro de Enseñanza, Investigación y Extensión en Producción Avícola CEIEPAv, Tláhuac 13300, Mexico; cuevasarturo@yahoo.com (A.C.-C.); avilaernesto@yahoo.com (E.Á.-G.); 5Departamento de Producción Avícola, Facultad de Medicina Veterinaria y Zootecnia, Universidad Michoacana de San Nicolás de Hidalgo, Santiago Tapia 403, Centro, Morelia 58000, Mexico; josearce_55@yahoo.com.mx; 6Departamento de Producción Agrícola y Animal, Universidad Autónoma Metropolitana, Calz. del Hueso 1100, Coapa, Villa Quietud, Coyoacán, Ciudad de México 04960, Mexico; dmartinez@correo.xoc.uam.mx

**Keywords:** β-mannanases, bacteriophage, enramycin, microbiome, broilers

## Abstract

**Simple Summary:**

For several years, antibiotic growth promoters (AGPs) have been used in poultry production; however, with the recent ban on the use of AGPs, several alternatives have been evaluated. In the present work, we evaluated the use of β-mannanases and bacteriophages as an alternative to AGPs. This study demonstrates that supplementation with β-mannanases, bacteriophages, or a mix of these two does not affect the productive performance in broilers fed corn–soybean meal. The mixture of β-mannanases and bacteriophages promoted the abundance of beneficial microorganisms in the cecum. These preliminary results suggest that β-mannanases and bacteriophages have potential as alternatives to AGPs in poultry production.

**Abstract:**

This study was conducted to evaluate the productive performance, intestinal health, and description of the cecum microbiota in broilers supplemented with β-mannanases (MNs) and bacteriophages (BPs). Six hundred one-day-old broilers were divided into four groups and fed one of the following diets: CON—corn–soybean meal + 10 ppm enramycin (ENR); MN: CON + 500 ppm MN; BP: CON + 500 ppm BP; MN + BP: BP + 500 ppm MN. The BP and MN factors showed similar performances to ENR. MN improved the concentration of IgA in the jejunum at 35 days of age. The morphometric index (IM) of the thymus increased by adding MN, while BP increased the liver and thymus IM. The histological analysis showed that BP and MN improved the intestinal morphology. MN + BP showed a tendency to decrease the abundance of *Proteobacteria* and increase the abundance of *Bacteroidetes*, indicating better microbiota function. In conclusion, our results demonstrate that the combination of MN + BP has potential in poultry nutrition; however, we highly recommend further experiments to confirm this hypothesis.

## 1. Introduction

Antibiotic growth promoters (AGPs) have been utilized as a part of regular practice in the poultry industry to improve performance and prevent disease [1]. However, the tendency worldwide is to reduce and avoid the use of AGPs in animal feed. Nowadays, there are different alternatives such as vaccination, probiotics, phytogenics, prebiotics and bacteriophages, organic acids, and enzymes [1]. The integrity of the intestine, along with the complex and diverse intestinal microbiota, has an important role in the absorption of nutrients, immune system development, and pathogen inhibition [2]. Factors, such as diet [3,4], age [5,6,7], health [8,9], environment [10], and feed additives, such as AGPs [11,12], have direct effects on the both the integrity and the microbiota in the gastrointestinal tract of metazoans.

Even though the use of AGPs improves weight gain and feed efficiency [12], it has been reported to cause a decrease in the population of several bacterial species, allowing the proliferation of antibiotic-resistant species that could affect the host and consumer health [11,13]. An alternative to AGPs is the use of bacteriophages [14,15,16]. It is well known that bacteriophages are an efficient treatment for bacterial diseases [17,18,19]. However, there is less information about the effect of bacteriophages on the performance, gut health, and microbiome of broiler chickens. On the other hand, the use of enzymes, such as β-mannanases, that have the capacity to hydrolyze antinutritional factors in feed grains, such as galactomannans, improves body weight gain and the feed conversion ratio [20,21].

To our knowledge, there is no information about the use of bacteriophages and β-mannanases in combination as an alternative to AGPs to improve performance or their effect on the gut morphology and cecum microbiota of broiler chickens fed corn soybean meal diets. Hence, the purpose of the present investigation was to evaluate the effect of bacteriophages alone or in combination with β-mannanases on performance parameters, intestinal morphometric analysis, and the microbiota and to compare these effects against those of the AGP enramycin in broiler chickens.

## 2. Materials and Methods

### 2.1. Facilities and Care of Experimental Animals

The experiment was carried out in accordance with Official Mexican Norm (NOM-033-SAG/ZOO-2014) guidelines for animal welfare, and experimental protocols were approved by the Institutional Animal Care and Use Committee of the College of Veterinary Medicine at the National Autonomous University of Mexico (CICUAE-FMVZ-UNAM MC-2017/1-14). The experiment was conducted in the facilities of the Center for Education, Research, and Extension in Poultry Production (CEIEPAv), National Autonomous University of Mexico (UNAM).

### 2.2. Experimental Design and Animal Management

A total of 600 one-day-old Ross 308^®^ broiler chickens (50:50 sex ratio) were distributed randomly to 24 pens. The broiler barn is an open-sided facility with a concrete floor housing 2.5 m^2^ pens equipped with individual feeders, individual in-line medicators and freshwater. The heat is provided via air heaters, and the facility has six fans for heat relief. Each pen contained wheat straw litter.

The experimental design comprised four treatments as follows:

Treatment 1 (CON): corn–soy meal + 10 ppm antibiotic growth promoter (Enramycine, Enradin^®^ F80, MSD Animal Health is a division of Merck & Co., Inc., Kenilworth, NJ, USA).

Treatment 2 (MN): as T1 + 500 ppm of β-mannanases (BM CIBENZA DE200^®^, Novus International, Inc., Saint Charles Missouri, MI, USA).

Treatment 3 (BP): corn–soy meal + 500 ppm of Bacteriophage Cocktail (BacterPhage C, CTCBIO, Inc., Seoul, Korea).

Treatment 4 (MN + BP): as BP + 500 ppm of β-mannanases (BM CIBENZA DE200^®^, Novus International, Inc., Saint Charles Missouri, MI, USA) + 500 ppm of Bacteriophage Cocktail (BacterPhage C, CTCBIO, Inc., Seoul, Korea).

BacterPhage C is composed of various bacteriophages targeting Salmonella Gallinarum, Salmonella Typhimurium, Salmonella Enteritidis, Salmonella Dublin, Salmonella derby, Staphylococcus aureus, Escherichia coli K99 and F41, and Clostridium perfringens type A and C [22].

Each treatment included 6 replicates of 25 chickens each. Broiler chicks were distributed in a completely randomized design with a 2 × 2 factorial arrangement. The first factor corresponded to Enramicyn or bacteriophage mix, and the second factor corresponded to feed with or without β-mannanase inclusion. The temperature during the experiment was lowered gradually from 32 at one day of age to 21 °C at 28 days of age. All diets were based on corn–soybean meal (Table 1); water and feed were provided ad libitum.

### 2.3. Productive Performance

Broilers and feed were weighed weekly, and the weight gain, feed intake, and feed conversion index were obtained. Mortality was recorded daily. Carcass yield was evaluated at 49 days.

### 2.4. Systemic Humoral Immune Response

To evaluate the systemic immune response at 10 days of age, broilers were simultaneously vaccinated with live-virus vaccine against Newcastle disease via the ocular route and a killed virus vaccine against Newcastle disease subcutaneously (La Sota^®^ Newcastle strain Laboratorios Avilab, S.A. de C.V. Porcicultores No. 80 Colonia Las Aguilillas 47,698 Tepatitlán de Morelos, Jal. and Newcastle Plus^®^, Laboratorios Avilab Tepatitlán de Morelos, Jal., respectively). At day 21, 2 mL of blood was taken from six chicks of each treatment (one per replication (*n* = 24)); sera were obtained and frozen at −20 °C in order to determine serum antibody titers specific for the ND virus through the hemagglutination inhibition test (Thayer and Beard, 1998).

### 2.5. Quantification of Intestinal Immunoglobulin A (IgA) Antibodies

To estimate the total (unspecific) IgA production in the jejunum epithelium, a commercial antigen capture ELISA chicken IgA quantitation kit (Bethyl Laboratories, Inc., Montgomery, TX, USA) was used following the manufacturer’s recommendations. At 21 days of age, 10 broilers per treatment were sacrificed by cervical dislocation and a 10 cm section was removed from the jejunum of each broiler. This was performed as previously reported [23].

### 2.6. Morphometric Index

Individual weighing of the liver and thymus of 40 broilers per treatment was carried out as previously reported [24].

### 2.7. Evaluation of Gut Morphology

At 35 days of age, the gut morphology of the duodenum was evaluated by measuring the villus length (VL), crypt depth (CD), and villus length/crypt ratio (VL/CD) [25]. Duodenum sections of 3 cm in length were obtained in 10% formalin for later analysis. The measurements were made in whole slides stained with hematoxylin and eosin (H&E) using a Leica DM500 photomicroscope (Leica Microsystems, Heerbrugg, Switzerland) with 4X objective and the help of the Leica LAS EZ software (Leica Application suite) version 3.3.0. For statistical analysis, 40 measurements from different fields of 12 samples per treatment were considered.

### 2.8. DNA Extraction, PCR, and Library Preparation for Sequencing

For the cecal DNA isolation, 200 µL of sample (*n* = 36) was extracted using the ZymoBIOMICS^TM^ DNA Miniprep kit (D4300 Zymo Research, Irvine, CA, USA), following the manufacturer’s instructions, followed by determination of the integrity, concentration, and purity. The DNA was stored at −80 °C until analysis.

The V4 region of the 16S rRNA gene from the genomic DNA of each of the 35 samples was amplified using the primers 515F [26] and 806R [27]. The library of amplicons for DNA sequencing was prepared according to the 16S Illumina PCR protocol described in the Earth Microbiome project (http://www.earthmicrobiome.org, accessed on 22 January 2019) with slight modifications [20]. In brief, the Q5^®^ High-Fidelity DNA Polymerase user guide protocol (New England Biolabs, Catalog No. M0491S) was used to conduct PCR in a 25 μL final reaction volume via 30 amplification cycles. The length of the amplified product was confirmed via 1% agarose gel electrophoresis, and equal amounts (~300 ng) of the amplicons from each sample as measured by Qubit dsDNA BR Assay Kit (ThermoFisher Scientific, Waltham, MA, USA, Catalog No. Q32850) were pooled together. The pooled amplicons were finally run on 1% agarose gel electrophoresis, purified using a Zymoclean Gel DNA Recovery Kit (Zymo Research, Catalog No. D4007), and sequenced via Illumina MiSeq paired-end 300-cycle options at Admera Health, LLC.

### 2.9. Amplicon Sequence Analysis

The Nebula cloud computing platform of the University of Arkansas was used to process the raw sequencing reads in QIIME 2 version 2018.8 utilizing the pipelines developed for paired-end data types [28]. In summary, the “demux emp-paired” method of the q2-demux plugin was used to demultiplex sequencing reads, followed by quality filtering and denoising with the “dada2 denoise-paired” method of q2-dada2 [29], available at QIIME 2. The truncation lengths of the forward and reverse reads were set to 220 and 200 bp, respectively, based on the quality score criteria (≥30). Taxonomic assignment was performed using a naïve Bayes classifier pretrained with Greengenes (version 13.8) 99% OTUs [30] and the q2-feature-classifier plugin, where the sequences were trimmed to include only the V4 region of the 16S rRNA gene region, the ends of which were defined by the 515F/806R primer pair. We detected the sequence reads assigned to chloroplasts and mitochondria, which were subsequently, removed using the taxonomy-based filtering option in QIIME2. The core-metrics-phylogenetic method at a sampling depth of 41,000 was used to analyze the alpha and beta diversity. The observed OTUs and Shannon indices were used to calculate alpha diversity, while the weighted UniFrac distance and unweighted UniFrac distance metrics were used for beta diversity analysis. All figures were created using ggplot2 packages in R [31]. Statistical differences among the treatment groups for different taxonomic groups were determined using the Kruskal–Wallis test followed by Wilcoxon for each pair comparison using JMP Genomics9. The significant differences in alpha diversity were calculated using the alpha-group-significance command of QIIME2, which is based on the Kruskal–Wallis test. Statistical differences in beta diversity among the groups were calculated by PERMANOVA [32] test using the beta-group-significance command of QIIME2 with the pairwise option. For both diversity analyses, the corrected *p*-values for multiple comparisons (q) were used to report significant differences between two groups, where the level of significance was set at q < 0.05.

### 2.10. Statistical Analysis

The results were analyzed using a completely randomized design with a 2 × 2 factorial arrangement treatment; one factor was the addition of bacteriophage or enramycin and the other factor was with or without β-mannanases. The antibody titer to ND was transformed to logarithm base 2. The results of the evaluated variables were analyzed using JMP^®^ computer software version 8. The multivariate variances in bacterial community composition were assessed in accordance with the guide to statistical analysis in microbial ecology (GUSTAME) [33]. Briefly, the vegan package in R 4.0.3 software was used to perform an analysis of the similarity (ANOSIM) among treatments. We performed PCoA analysis at the phylum, family, and genus levels using the vegan package in R 4.0.3 software. The relative abundances of microbial communities and alpha diversity variables (Chao 1 index and Shannon index) were analyzed via the Student’s *t*-test using R 4.0.3 software.

## 3. Results

### 3.1. Productive Performance

Table 2 shows the results of the effect on the performance of broilers at 49 days of age when fed corn–soybean meal diets with added enramycin, bacteriophages, and β-mannanases. No significant differences in terms of weight gain, feed consumption, feed conversion ratio, carcass yield, or mortality were observed (*p* > 0.05) between the control and experimental groups (Table 2).

### 3.2. Systemic Humoral Immune Response

The results of the effect on the morphometric index (day 49) and IgA jejunum concentration (day 35) in broilers fed corn–soybean meal diets with added enramycin, bacteriophages, and β-mannanases are summarized in Table 3. A significant increase (*p* < 0.01) in the jejunum epithelium IgA due to the β-mannanases was observed (Table 3).

### 3.3. Morphometric Index

Table 3 shows the results of the effect on the morphometric index (day 49) of broilers fed corn–soybean meal diets with added enramycin, bacteriophages, and β-mannanases. In the present study, the bacteriophage factor showed an increase (*p* < 0.01) in the liver and thymus size (Table 3).

### 3.4. Evaluation of Gut Morphology

The results of the morphometric evaluation of the duodenum in broilers fed corn–soybean meal diets with added enramycin, bacteriophages, and β-mannanases at 35 days are summarized in Table 4. A significant increase in villus length (VL) (*p* < 0.01) was observed with the bacteriophage factor. Crypt depth (CD) was reduced (*p* < 0.01) with β-mannanase treatment. However, the VL/CD ratio increased (*p* < 0.05) with β-mannanase supplementation (Table 4).

### 3.5. Cecum Microbiota Analysis

#### 3.5.1. Summary of DNA Sequence Data

Summarization of the feature table resulted in a total of 3,096,219 sequence reads from the 35 samples, ranging from 41,108 to 157,031 reads per sample. The median and mean ± SE numbers of reads per sample were 89,479 and 88,463.4 ± 4959.72, respectively. In addition, there were altogether 1417 unique features (amplicon sequence variants) from all samples.

#### 3.5.2. Alpha Diversity Analysis

To determine whether there were any differences in alpha diversity (diversity within the community) among the treatment groups, we performed an alpha diversity analysis. As shown in Figure 1, there were numerical differences in the alpha diversity among the groups when measured by the observed OTUs (Figure 1A) and Shannon index (Figure 1B), but none of them were statistically significant (Kruskal–Wallis test; *p* < 0.05).

#### 3.5.3. Beta Diversity Analysis

We also performed beta diversity analysis to determine the differences in community structure among the treatment groups using four different metrics (weighted UniFrac, unweighted UniFrac, Jaccard, and Bray–Curtis) for the measurement of beta diversity. However, none of them showed statistical differences among the four treatment groups at q < 005, which is also reflected in the PCoA plots generated based on (1) the weighted UniFrac distance metric (Figure 2A) and (2) the unweighted UniFrac distance metric (Figure 2B).

#### 3.5.4. Taxonomic Assignments

Taxonomic Assignment at the Phylum Level

The ANOSIM (*r* = −0.023, *p* = 0.711) and PCoA analyses demonstrated that there were no significant differences among the groups at the phylum level (Figure 3A). There were 16 identified phyla of cecal bacteria, and only those with relative abundance exceeding 0.1% of the total are listed (Appendix A). The predominant phylum in all treatments was Firmicutes (71.67, 67.10, 79.43, and 76.90% for the CON, MN, BP, and MN + BP diets, respectively), followed by Bacteroidetes (CON: 10.06%, MN: 8.55%, BP: 8.29%, and MN + BP: 12.12%), Proteobacteria (CON: 6.97%, MN: 14.53%, BP: 7.12%, and MN + BP: 2.16%), and Actinobacteria (CON: 7.74%, MN: 8.15%, BP: 2.49%, and MN + BP: 7.29%) (Figure 3B).

Taxonomic Assignment at the Family Level

The ANOSIM (*r* = −0.072, *p* = 0.983) and PCoAs demonstrated that there were no significant differences among the groups at the family level (Figure 4A). There were 76 identified families of cecal bacteria, and only those with relative abundance exceeding 0.1% of the total are listed (Appendix A). The predominant family in all treatments was Ruminococcaceae (CON: 34.33%, MN: 31.08%, BP: 37.89%, and MN + BP: 40.76%)*,* followed by Lachnospiraceae CON: 21.34%, MN: 18.20%, BP: 23.62%, and MN + BP: 20.47%), *Bifidobacteriaceae* (CON: 6.94%, MN: 7.90%, BP: 2.15%, and MN + BP: 7.03%), Veillonellaceae (CON: 3.53%, MN: 10.55%, BP: 5.38%, and MN + BP: 7.28%), and Enterobacteriaceae (CON: 3.49%, MN: 9.77%, BP: 4.60%, and MN + BP: 1.28%) (Figure 4B).

Taxonomic Assignment at the Genus Level

The ANOSIM (*r* = −0.027, *p* = 0.698) and PCoAs demonstrated that there were no significant differences among the groups at the genus level (Figure 5A). There were 129 identified genera of cecal bacteria, and only those with relative abundance exceeding 0.1% of the total are listed (Appendix A). The predominant genus in all treatments was Faecalibacterium (CON: 11.14%, MN: 12.31%, BP: 11.42%, and MN + BP: 22.32%) followed by Oscillospira (CON: 7.40%, MN: 5.22%, BP: 7.94%, and MN + BP: 4.27%), Bifidobacterium (CON: 6.97%, MN: 7.91%, BP: 2.15%, and MN + BP: 7.04%), Ruminococcus (CON: 3.82%, MN: 3.16%, BP: 2.73%, and MN + BP: 2.73%), a non-assigned genus of Lachnospiraceae (CON: 9.97%, MN: 6.15%, BP: 8.49%, and MN + BP: 8.96%)*,* a non-assigned genus of Ruminococcaceae (CON: 9.80%, MN: 7.62%, BP: 9.43%, and MN + BP: 9.11%)*,* and a non-assigned genus of Rikenellaceae (CON: 6.26%, MN: 5.54%, BP: 2.87%, and MN + BP: 9.01%) (Figure 5B).

## 4. Discussion

### 4.1. Productive Performance

The use of growth factors in poultry has been a common practice to increase productivity and efficiency; however, in recent years due to the growing concern of the development of microbial resistance, its use is under regulation [34,35].

New strategies, such as feed enzymes and bacteriophage mixes, have been evaluated as an alternative to antibiotic growth promoters; thus, it has become a crucial challenge in the poultry industry to elucidate their mode of action.

It has been reported that the inclusion of β-mannanases in poultry diets enhances performance and the feed conversion ratio [20,36]. On the other hand, bacteriophages are bacterial viruses with high specificity, which represents an advantage over antibiotics [37].

It is well know that supplementation with bacteriophages improves the health status of farm animals [37]; however, its effect on growth performance has shown inconsistent results; for instance, in calves, supplementation with bacteriophages did not increase growth performance but improved the health status of the animal [22]. In weaned pigs, it has been demonstrated that the use of bacteriophages enhances growth performance [38].

In the present study, we did not observe significant differences in broiler growth performance with growth promoter or with the addition of β-mannanases or bacteriophages or the mixture of the latter; our results agree with previous reports where it was shown that supplementing poultry with bacteriophages does not enhances growth performance [39]. To our knowledge, there are few studies on the combined used of β-mannanases and bacteriophages. In calves, it was demonstrated that the combination of β-mannanases and bacteriophages does not improve growth performance but enhances the inflammatory response [22].

Thus, even though we did not observe differences in the growth performance between treatments, our results suggest that the bacteriophage mix, β-mannanases, or a combination of these could be used as an alternative to growth factors.

### 4.2. Humoral Immune Response

The increase in secretory IgA could be due to the binding of mannose receptors to mananooligosaccharides (MOS), which are generated by the hydrolytic activity of the β- mannanases. It has been established that MOS are immunostimulants that affect the immune response, and their mechanism of action can be explained by two main processes: (a) MOS are recognized as PAMPs (pathogen-associated molecular patterns) due to the stimulation of the lymphoid tissue associated with the intestine firing the immune response [40]; (b) MOS stimulate the synthesis of mannose-binding proteins, causing an increase in phagocytosis and enhancement of the immune response [41]. Thus, the increase in IgA in the present study could be attributed to the action of the β-mannanases.

The gut morphology is an important indicator of intestinal health. In this study, β-mannanases did not interfere with the morphometric indices of the organs evaluated, in accordance with previous reports [42,43]. Dietary bacteriophages increased the morphometric index of the liver and thymus as previously reported by Wang [44], but this effect is still poorly documented and understood.

Interestingly, at day 35, a significant increase was observed in the VL/CD ratio in the duodenum in chickens that received β-mannanases; it has been demonstrated that a higher VL/CD ratio is associated with improved nutrient absorption [45]. The changes observed at day 35 could be explained through the action of β-mannanases, which hydrolyzed β-mannans into MOS [46]. Previous reports demonstrated that supplementation with 0.5% MOS improved villus height in 14 day old broilers [47].

### 4.3. Cecum Microbiota Analysis

Chicken gut microbiota have been widely studied; our results showed, as previously reported, that the main phyla in the chicken cecum were *Bacteroidetes, Firmicutes*, and *Proteobacteria* [48]. Even though it was not significantly different (*p* = 0.06), we observed that feeding the animals with a combination of β-mannanases and bacteriophages (MN + BP) caused a decrease in the abundance of the *Proteobacteria* phylum in comparison with that in the CON, MN, and BP groups. Several pathogens, such as *Escherichia coli*, *Campylobacter jejuni*, *Klebsiella pneumoniae*, *Salmonella typhimurium*, and *Yersinia enterocolitica*, are included within this phylum [29]. In humans and in murine models, an increase in the *Proteobacteria* phylum is related to dysbiosis and the prevalence of diseases such as metabolic disorders, inflammation, and cancer [48]; in chickens, the presence of *Proteobacteria*, particularly an increase in the *Enterobacteriaceae* family, causes a decrease in performance parameters [39].

The changes observed in the abundance of the phylum *Proteobacteria* and the family *Enterobacteriaceae* when feeding the animals with MN + BP could be explained by the mechanism of action of both additives; β-mannanases are enzymes that hydrolyze β-mannans and yield mannan oligosaccharides (MOS), which bind to live pathogenic bacteria *E. coli* and *Salmonella*, avoiding their proliferation in the intestine [45]; meanwhile, the bacteriophage mix used in the present study is specific for *Salmonella* species, *Staphylococcus aureus*, *Escherichia coli* K99 and F41, and *Clostridium perfringens* type A and C [22]. This indicates that the combined use of β-mannanases and bacteriophages may be a good strategy to improve animal health without affecting performance.

Even though it was not statistically significant, we observed an increase in the relative abundance of the Bacteroidetes phylum and unclassified Rikenellaceae in broilers fed MN + BP for 49 days. The Bacteroidetes phylum is composed of bacteria that ferment polysaccharides and indigestible carbohydrates [49], and its presence in chickens is associated with fat accumulation. Within the Bacteroidetes phylum, two genera have been associated with the biosynthesis of fatty acids and lipid metabolism: Alistipes and the unclassified Rikenellaceae [50]. Thus, the combination of β-mannanases and bacteriophages could regulate lipid metabolism; however, it is necessary to conduct further studies to evaluate this mechanism.

The abundance of the Ruminococcaceae and Lachnospiraceae families is associated with gut health [43]. Feeding animals with MN + BP and BP caused a non-significant increase in both families when compared to the group supplemented with enramycin (CON), suggesting that the use of β-mannanases and bacteriophages has no effect on gut health.

At the genus level, we observed a non-significant decrease (*p* = 0.07) of Brachybacterium in the BP and MN + BP groups in comparison with the CON and MN groups. In humans, the presence of Brachybacterium causes bloodstream infections [51], and in dairy cows Brachybacterium is related to mastitis development [52]. In poultry, an increase in the relative abundance of Brachybacterium has been associated with low poultry performance, and it could be related to disease in chickens [53]. results suggest the use of β-mannanases and bacteriophages may improve animal health.

Oscillospira is a butyrate-producing microorganism [54]. Butyrate is a volatile fatty acid (VFA) produced by the gut microbiota; in humans, it has been reported that butyrate helps in controlling metabolic syndrome through the improvement of glucose uptake in adipose tissue, enhancing insulin signaling in liver and increasing the secretion of glucagon like peptide 1 (GLP-1) [55]; in farm animals, it has been reported that butyrate promotes body weight and composition [56]. The animals fed BP showed a higher abundance (*p* = 0.06) of Oscillospira in comparison to the CON, BP, and MN + BP groups, indicating that the use of bacteriophages may promote body weight.

In this study, we analyzed the effect of a bacteriophages with high affinity to *Salmonella* spp. Even though we did not observe differences in *Salmonella* spp., our results demonstrated that the use of BacterPhage C^®^ enhanced the abundance of beneficial microorganisms in the cecum.

The use of bacteriophages in poultry has gained importance in recent years, and the most commonly used bacteriophages are targeted to several pathogen microorganisms of interest such as *Campylobacter jejuni*, *Escherichia coli*, *Listeria monocytogenes*, and *Staphylococcus aureus.* Recent studies demonstrated that the use of a bacteriophage cocktail may reduce the abundance of pathogens in poultry [37]. Further studies are necessary to understand the mechanism of action of a bacteriophage cocktail on the productive performance and cecum microbiota abundance.

## 5. Conclusions

The performance parameters in chickens fed bacteriophages and β-mannanases were similar to those in chickens fed enramycin. β-Mannanases increased IgA levels in the jejunum at 35 days and improved the morphometric index of the thymus, while bacteriophages improved the morphometric indices of the liver and thymus. The combination of bacteriophages and β-mannanases improved gut morphology and tended to reduce Proteobacteria and enhance Bacteroidetes. The present work was an explorative study that allowed us to elucidate the possible mechanism of action of the inclusion of β-mannanases and/or bacteriophages on the cecum microbiota; even though we did not observe significant changes in the cecum microbial population, we observed some tendencies that may indicate that the use of these additives could be a good alternative to improve performance in chickens without affecting the cecum microbiota. Further studies are necessary to understand how gut health and the cecum microbiota adapt to these additives.

## Figures and Tables

**Figure 1 animals-12-00169-f001:**
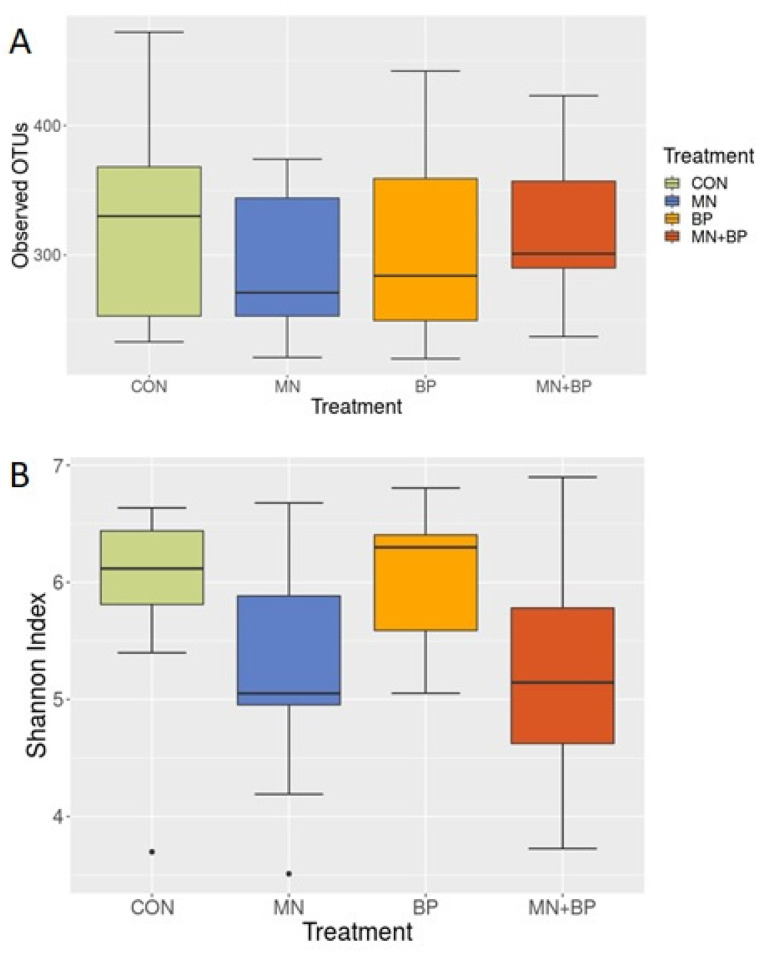
Observed OTUs (**A**) and Shannon index (**B**) of the cecal content of broilers fed with CON, MN, BP, and MN + BP.

**Figure 2 animals-12-00169-f002:**
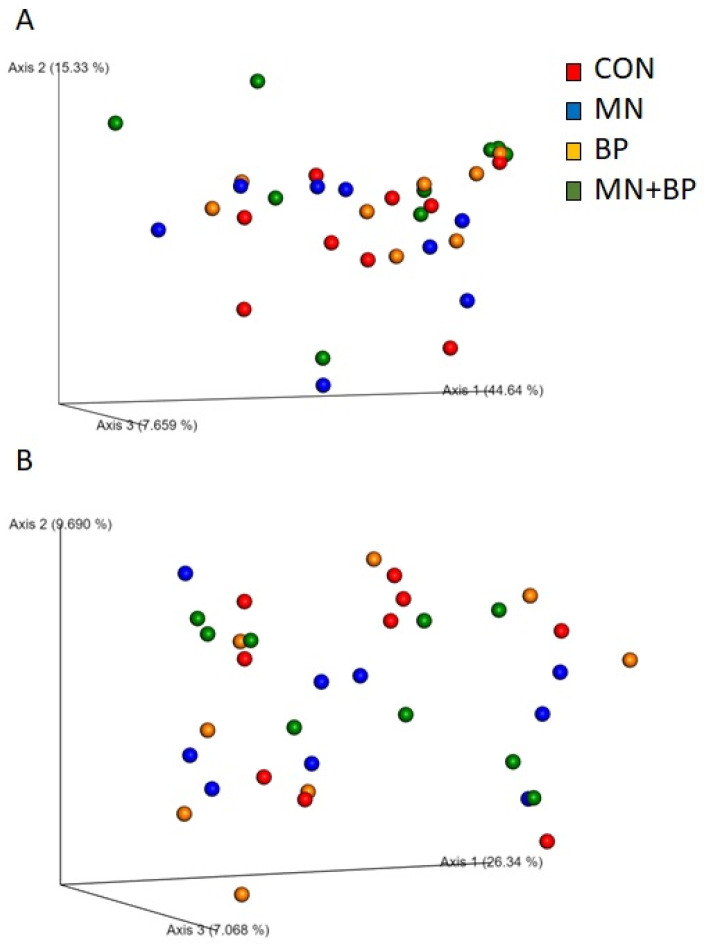
The weighted UniFrac distance metric (**A**) and unweighted UniFrac distance metric (**B**) of the cecal content of broilers fed with CON, MN, BP, and MN + BP.

**Figure 3 animals-12-00169-f003:**
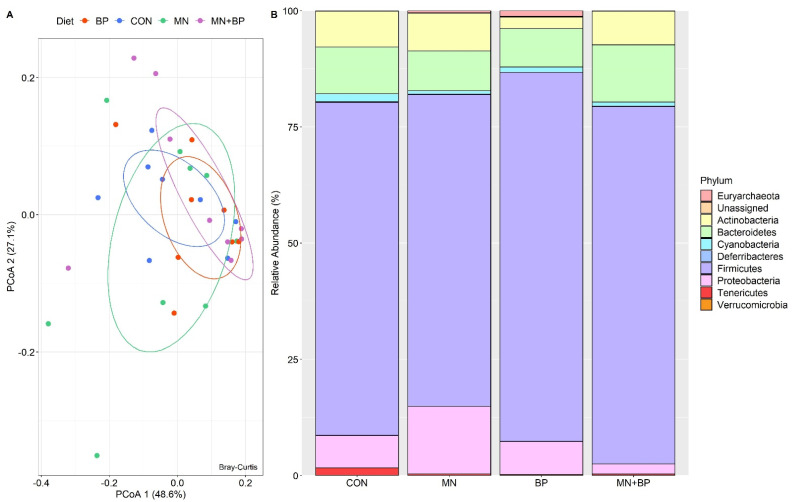
Bacterial composition at the phylum level in cecal content of broilers fed with CON, MN, BP, and MN + BP: (**A**) principal component analyses (PCoAs); (**B**) relative abundance (%) of bacteria at the phylum level.

**Figure 4 animals-12-00169-f004:**
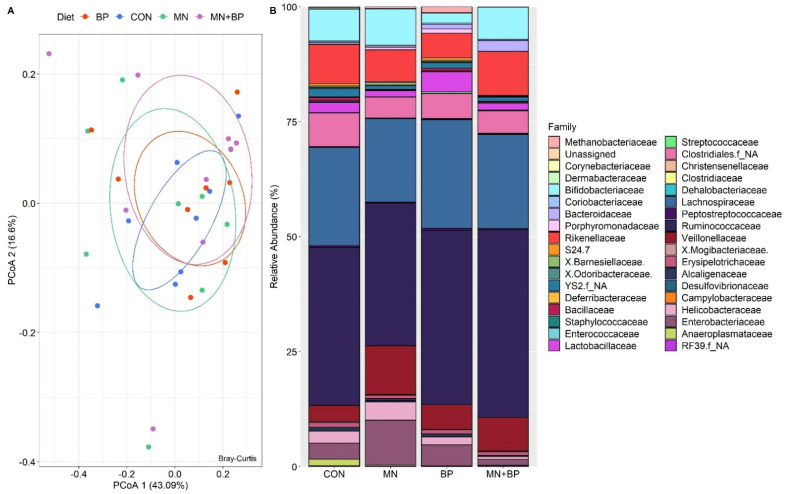
Bacterial composition at the family level in the cecal content of broilers fed with CON, MN, BP, and MN + BP: (**A**) principal component analyses (PCoAs); (**B**) relative abundance (%) of bacteria at the family level.

**Figure 5 animals-12-00169-f005:**
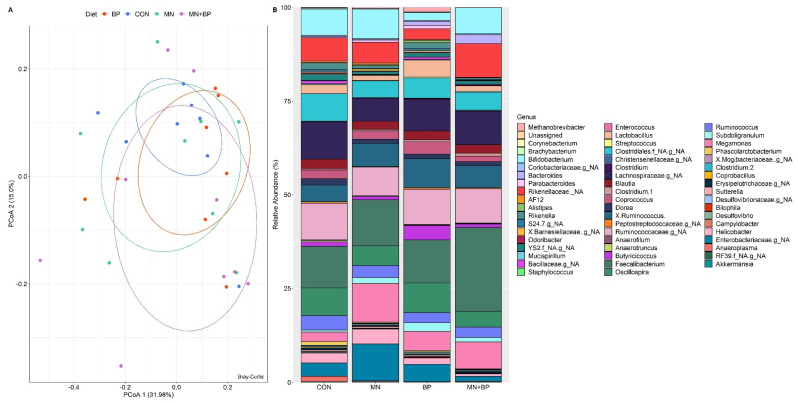
Bacterial composition at the genus level in the cecal content of broilers fed with CON, MN, BP, and MN + BP: (**A**) principal component analyses (PCoAs); (**B**) relative abundance (%) of bacteria at the genus level.

**Table 1 animals-12-00169-t001:** Feed composition of the experimental diets.

Ingredients (g/kg)	Starter (1–21 d)	Finisher (22–49 d)
Corn	570.405	539.30
Soybean meal	370.925	289
Vegetable oil	18.425	20.4
Calcium carbonate	14.75	14.5
Orthophosphate	10.55	10.1
Salt	3.5	3.5
DL-methionine	3.12	3.5
L-lysine HCl	2.85	4
L-threonine	0.725	-
Mineral and Vitamin premix *	3.5	3.5
Phytase	0.55	0.55
Avelut^®^ (pigment)	-	6.0
Coccidiostat	0.5	5.0
Choline chloride 60%	0.05	0.5
Antioxidant **	0.15	0.15
Total	1000	1000
Calculated composition		
Metabolizable energy Kcal/kg	3010	3200
Crude protein (%)	22	19
Digestible lysine (%)	1.44	0.94
Digestible Met + Cys (%)	0.9	0.73
Total calcium (%)	0.96	0.85
Available phosphorus (%)	0.48	0.42

* Mineral and vitamin premix provided: vitamin A 12,000,000 IU, vitamin D3 2,500,000 IUP, vitamin E 15,000 IU, vitamin K3 2000 mg/kg, vitamin B1 2250 mg/kg, vitamin B2 7500 mg/kg, vitamin B3 45,000 mg/kg, vitamin B5 12,500 mg/kg, vitamin B6 3500 mg/kg, vitamin B12 20 mg/kg, folic acid 1500 mg/kg, biotin 125 mg/kg, iodine 300 mg/kg, selenium 200 mg/kg, cobalt 200 mg/kg, iron 50,000 mg/kg, copper 12,000 mg/kg, zinc 50,000 mg/kg, manganese 110,000 mg/kg. ** BHT (1.2%) and BHQ (9%).

**Table 2 animals-12-00169-t002:** Effect on performance on broilers at 49 days age fed with corn–soybean meal diets added with growth promoters and β-mannanases.

Treatment	Weight Gain (g)	Feed Consumption (g)	Feed Conversion Ratio (kg/kg)	Carcass Yield (%)	Mortality (%)
Growth Promoter					
AGP	2873	5410	1.89	69.7 ^a^	5.1 ^a^
BF	2860	5459	1.91	69.2 ^a^	4.7 ^a^
β-Mannanases					
-	2898	5471	1.90	69.7 ^a^	4.1 ^a^
+	2835	5398	1.91	69.3 ^a^	5.6 ^a^
	*p*-Value				
Growth Promoter	0.91	0.67	0.60	0.14	0.9
β-Mannanases	0.58	0.52	0.81	0.18	0.4
Interaction	0.80	0.77	0.54	0.97	-
SEM	270	2750	0.10	1.87	3.4

Two factors were examined and compared in this trial: (1) AGP—enramycin or BF—bacteriophages; (2) with and without β-mannanases. The absence of literals between the means of each column indicates that there were no significant statistical differences (*p* < 0.05). SEM = standard error of the mean. Presence of literal superscript (a) indicates statistically significant differences.

**Table 3 animals-12-00169-t003:** Effect on the morphometric index of liver and thymus at day 49 and IgA jejunum concentration at day 35 on broilers in corn–soybean meal diets added with growth promoters and β-mannanases.

Treatment	Morphometric Index Day 49		Intestinal IgA (ng/mL) d 35
	Liver	Thymus	
Growth Promoter			
AGP	1.89 ^b^	0.18 ^b^	136.5 ^b^
BF	2.01 ^a^	0.22 ^a^	146.0 ^a^
β-Mannanases			
-	1.97 ^a^	0.19 ^b^	122.5 ^a^
+	1.93 ^a^	0.21 ^b^	160.4 ^b^
*p*-Value			
Growth Promoter	0.002	0.002	0.47
β-Mannanases	0.26	0.10	0.01
Interaction	0.17	0.12	0.08
SEM	0.24	0.08	51.60

Two factors were examined and compared throughout the study: (1) AGP—enramycin or BF—bacteriophages; (2) with and without β-mannanases. The absence of literals between the means of each column indicates that there were no significant statistical differences (*p* < 0.05). SEM = standard error of the mean. Presence of literal superscripts (a, b) indicate statistically significant differences.

**Table 4 animals-12-00169-t004:** Histological evaluation of the duodenum on broilers in corn–soy diets added with growth promoters and β-mannanases.

Treatment	Day 35		
	VL	CD	VL/CD Ratio
Growth Promoter			
AGP	2187.5 ^a^	286.6 ^a^	8.0 ^a^
BF	2304.0 ^b^	276.7 ^a^	8.3 ^a^
β-Mannanases			
-	2230.6 ^b^	291.6 ^a^	8.0 ^a^
+	2266.0 ^b^	271.3 ^b^	8.4 ^b^
	*p*-Value		
Growth Promoter	<0.0001	0.13	0.17
β-Mannanases	0.23	0.002	0.04
Interaction	0.8	<0.0001	0.0002
SEM	146.1	37.5	1.3

Two factors were examined and compared throughout the study: (1) AGP—enramycin or BF—bacteriophages; (2) with and without β-mannanases. The absence of literals between the means of each column indicates that there were no significant statistical differences (*p* < 0.05). SEM = standard error of the mean. VL = villus length; CD = crypt deep; VL/CD ratio = villus length/deep crypt. Presence of literal superscripts (a, b) indicate statistically significant differences.

## Data Availability

The raw data supporting the conclusions of this manuscript will be made available by the authors, without undue reservation, to any qualified researcher.

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
