# Peer review of "Productive Performance and Cecum Microbiota Analysis of Broiler Chickens Supplemented with β-Mannanases and Bacteriophages—A Pilot Study"

_animals, 2022, doi:10.3390/ani12020169_

Round 1

Reviewer 1 Report

This study evaluates the use of beta-mannanase and bacteriophage, alone or in combination, as alternatives to traditional antibiotic growth promoters such as enramycin. The impact of these on intestinal health, performance, and the gastrointestinal microbiota were evaluated. There was no significant difference in the composition of the gastrointestinal microbiota between treatment groups, but there were differences in intestinal morphology and local immune responses between groups. Both beta-mannanase and bacteriophage appear to be promising alternatives.

Introduction

  • Line 68 – change the word microbiome to microbiota
  • Line 81 – should this be ‘floor pens' instead of ‘floors’?
  • Line 81 – what do you mean by ‘natural environment’? Please clarify this in the manuscript
  • Line 99 – change the word ‘repetitions’ to replicates

Materials and methods

  • Lines 185 to 194 – the description of your statistical analyses is vague. More details are needed in regards to analysis of the production metrics such as weight gain, FCR etc., and antibody titres
  • I would amend most of your acronyms by swapping the first and second letters. For example, VL seems like a more intuitive acronym for vilus length than LV, and CD seems like a more intuitive acronym for crypt depth than DC
  • Did you consider any other ways of monitoring the impact of each treatment on the microbial population of the gut? For example, it looks like you’ve provided a cocktail of bacteriophage targeting various Salmonella species. Did you consider using qPCR to monitor Salmonella populations?

Results

  • Table 6 – the p-value for Dermabacteriaceae is significant (p = 0.04). Should this have been 0.40?
  • Table 7 – there are several p-values that are approaching significant (i.e. just above 0.05) – these taxa might be worth discussing in the Discussion

Discussion

  • The Discussion needs further work
  • The first paragraph is disjointed and could be expanded
  • The “Productive performance” section of the Discussion needs to be extended and discussed in relation to other studies. At present this section is a repetition of material from the Results section
  • Your discussion of the impact of each treatment on gut morphology needs to be expanded

Author Response

Reviewer 1

This study evaluates the use of beta-mannanase and bacteriophage, alone or in combination, as alternatives to traditional antibiotic growth promoters such as enramycin. The impact of these on intestinal health, performance, and the gastrointestinal microbiota were evaluated. There was no significant difference in the composition of the gastrointestinal microbiota between treatment groups, but there were differences in intestinal morphology and local immune responses between groups. Both beta-mannanase and bacteriophage appear to be promising alternatives.

Dear Reviewer, #1, thank you very much for the time you have spent on reviewing our manuscript. Your comments are very valuable and helpful for revising our paper and guiding our research. We have studied those comments carefully and have made corrections, which we hope to meet with the approval. Revised portion in the new version were included and are highlighted in yellow in the reviewed manuscript. The following is our point-by-point response to reviewers’ comments:

Introduction

  • Line 68 – change the word microbiome to microbiota
  • Response: Line 68 the word microbiome has been changed to microbiota. Corrections made, thank you.
  • Line 81 – should this be ‘floor pens' instead of ‘floors’?
  • Response: Thank you for the observation, Line 81 the word floors has been changed to floor pens
  • Line 81 – what do you mean by ‘natural environment’? Please clarify this in the manuscript
  • Suggestion accepted and sentence was clarified.  Thank you.
  • Line 99 – change the word ‘repetitions’ to replicates
  • Response: Line 99 the word repetitions has been changed to replicates. Corrections made, thank you.
  •  

Materials and methods

  • Lines 185 to 194 – the description of your statistical analyses is vague. More details are needed in regard to analysis of the production metrics such as weight gain, FCR etc., and antibody titres

Response. This suggestion was made in the original paper. Thank you.

  • I would amend most of your acronyms by swapping the first and second letters. For example, VL seems like a more intuitive acronym for vilus length than LV, and CD seems like a more intuitive acronym for crypt depth than DC
  • Response: Thank you for the observation, we changed the acronyms
  • Did you consider any other ways of monitoring the impact of each treatment on the microbial population of the gut? For example, it looks like you’ve provided a cocktail of bacteriophage targeting various Salmonella species. Did you consider using qPCR to monitor Salmonella populations?
  • Response: The authors appreciate this excellent suggestion; in futher research in this area, it really would be interesting. We hadn't thought about it before.Thank you.

Results

  • Table 6 – the p-value for Dermabacteriaceae is significant (p = 0.04). Should this have been 0.40?
  • Response: Thank you for the observation, we corrected in original paper the error in the p value for Dermabacteriaceae.
  • Table 7 – there are several p-values that are approaching significant (i.e. just above 0.05) – these taxa might be worth discussing in the Discussion
  • Response: We included in the discussion section in original paper the taxa with a p value above 0.05. Thank you.

Discussion

  • The Discussion needs further work. We heeded this suggestion in the original text. Thank you.
  • The first paragraph is disjointed and could be expanded
  • The “Productive performance” section of the Discussion needs to be extended and discussed in relation to other studies. At present this section is a repetition of material from the Results section
  • Your discussion of the impact of each treatment on gut morphology needs to be expanded
  • Response, the authors deepen the discussion of the results, include references, and compare the results obtained with current literature in original paper.

Reviewer 2 Report

The manuscript describes a study to evaluate the productive performance and intestinal health in broilers supplemented with β-mannanases (MN) and bacteriophages (BP). Additionally, metagenomic analyzes of the broilers cecum microbiotas were carried out and the results were evaluated. In my opinion, the manuscript presents some interesting results. However I think the authors need to prepare better the whole manuscript.

In my overall analysis of the manuscript it seems the authors planed to evaluate the broilers overall performance using MN and BP with traditional tools and they included an additional metagenomic analysis of the cecum microbiota. Additionaly, the authors have not studied the intestinal health and "description" is not a proper word form the microbiota analysis. So my first suggestion is in the Title: the two main topics in the Results (“Productive performance and cecum microbiota analysis”) are more properly related to the whole study and they could be joined to construct a more proper title.

In the analysis of the different sections, the Introduction seems Ok and it presents a good bibliographic base on the main subject of the article. But Materials and Methods should be improved. This section is very itenized and some topics could be joined. Some important information is also lacking, as for example in the topic 2.2: it is not clear the main indicative use of the Bacteriophage Cocktail (BF Bacteriophage Plus). Is it recommended to control only the reported serovars (Gallinarum, Typhimuriun and Enteritidis)? What is the specific product? Please take a look in another study published in Animals (https://doi.org/10.3390/ani11020372 ) that describes more properly the product (including a reference with the number of the patent).

In the Results section, the manuscript presents many Figures and Tables without adding important information. Tables 6 and 7, for example, should be removed since the results are already presented in the Figures 4 and 5. I recommend the authors revise all this section, removing duplicate information in the Tables and Figures. And all removed material should be provided as supplementary.

Finally, I think the authors have not discussed properly the main results. There are few referred articles, including one topic (Productive Performance) without any reference. In addition, the authors have discussed the composition of the microbiota, but they have not compared with the bacteriophage used. Which bacteriophage products would they recommend using for other pathogenic bacterial species (other than Salmonella)? Therefore I suggest the authors to review the Discussion too.  The Conclusion should also be reviewed after the corrections in the whole manuscript.

Author Response

Response to Reviewer 2 Comments

Reviewer 2

The manuscript describes a study to evaluate the productive performance and intestinal health in broilers supplemented with β-mannanases (MN) and bacteriophages (BP). Additionally, metagenomic analyzes of the broilers cecum microbiotas were carried out and the results were evaluated. In my opinion, the manuscript presents some interesting results. However I think the authors need to prepare better the whole manuscript.

 Dear Reviewer, #2, thank you very much for the time you have spent on reviewing our manuscript. Your comments are very valuable and helpful for revising our paper and guiding our research. We have studied those comments carefully and have made corrections, which we hope to meet with the approval. Revised portion in the new version were included and are highlighted in yellow in the reviewed manuscript. The following is our point-by-point response to reviewers’ comments:

In my overall analysis of the manuscript it seems the authors planed to evaluate the broilers overall performance using MN and BP with traditional tools and they included an additional metagenomic analysis of the cecum microbiota. Additionaly, the authors have not studied the intestinal health and "description" is not a proper word form the microbiota analysis. So my first suggestion is in the Title: the two main topics in the Results (“Productive performance and cecum microbiota analysis”) are more properly related to the whole study and they could be joined to construct a more proper title.

Response: the authors analyze your excellent suggestion and accept it. Thank you.

In the analysis of the different sections, the Introduction seems Ok and it presents a good bibliographic base on the main subject of the article. But Materials and Methods should be improved. This section is very itenized and some topics could be joined. Some important information is also lacking, as for example in the topic 2.2: it is not clear the main indicative use of the Bacteriophage Cocktail (BF Bacteriophage Plus). Is it recommended to control only the reported serovars (Gallinarum, Typhimuriun and Enteritidis)? What is the specific product? Please take a look in another study published in Animals (https://doi.org/10.3390/ani11020372 ) that describes more properly the product (including a reference with the number of the patent).

 Response: The authors include the information of the product according to their suggestion in the original text. Thank you.

In the Results section, the manuscript presents many Figures and Tables without adding important information. Tables 6 and 7, for example, should be removed since the results are already presented in the Figures 4 and 5. I recommend the authors revise all this section, removing duplicate information in the Tables and Figures. And all removed material should be provided as supplementary.

  Response: the authors analyzed your valuable suggestion and eliminated tables 6 and 7 in the original text. Thank you.

Finally, I think the authors have not discussed properly the main results. There are few referred articles, including one topic (Productive Performance) without any reference. In addition, the authors have discussed the composition of the microbiota, but they have not compared with the bacteriophage used. Which bacteriophage products would they recommend using for other pathogenic bacterial species (other than Salmonella)? Therefore I suggest the authors to review the Discussion too.  The Conclusion should also be reviewed after the corrections in the whole manuscript.

Response: the authors accept your excellent suggestion, and this section has been rewritten in the original paper.

Round 2

Reviewer 2 Report

The manuscript describes a study to evaluate the productive performance and intestinal health in broilers supplemented with β-mannanases (MN) and bacteriophages (BP). Additionally, metagenomic analyzes of the broilers cecum microbiotas were carried out and the results were evaluated.

As I commented before, the manuscript is interesting and it presents some important findings. In this new version (R1), the authors proposed a more proper Title and performed the necessary modifications in the Materials and Methods as well as in the Results. More importantly, they improve the Discussion including more references as suggested. Therefore I think the manuscript can be accepted.